# Quorum Sensing in Oral Biofilms: Influence on Host Cells

**DOI:** 10.3390/microorganisms11071688

**Published:** 2023-06-28

**Authors:** Malee Nagi, Iain L. C. Chapple, Praveen Sharma, Sarah A. Kuehne, Josefine Hirschfeld

**Affiliations:** 1Oral Microbiology Group, School of Dentistry, Institute of Microbiology and Infection, College of Medical and Dental Sciences, University of Birmingham, Birmingham B5 7EG, UK; mxn627@student.bham.ac.uk (M.N.); s.a.kuehne@bham.ac.uk (S.A.K.); 2Periodontal Research Group, School of Dentistry, College of Medical and Dental Sciences, University of Birmingham, Birmingham B5 7EG, UK; i.l.c.chapple@bham.ac.uk (I.L.C.C.); p.sharma@bham.ac.uk (P.S.); 3NIHR Birmingham Biomedical Research Centre, Birmingham B5 7EG, UK

**Keywords:** quorum sensing, autoinducer-2 (AI-2), *N*-acyl homoserine lactones (AHL), periodontal diseases, oral biofilm

## Abstract

Quorum sensing molecules (QSMs) in the oral cavity regulate biofilm formation, the acquisition of iron, stress responses, and the expression of virulence factors. However, knowledge of the direct QSM–host interactions in the oral environment is limited, although their understanding could provide greater insight into the cross-kingdom communication occurring during oral disease development. This review aims to explore the literature on oral QSM–host interactions and to highlight areas of advancement in this field. The studies included in this review encompass an array of cell types and oral QSMs, with particular emphasis on immune cells and their relationship to periodontal diseases. It can be inferred from the current literature that QSMs are utilised by host cells to detect bacterial presence and, in the majority of cases, elicit an immune response towards the environmental QSMs. This may provide a base to target QSMs as a novel treatment of oral diseases. However, *N*-acyl homoserine lactone (AHL) detection methods remain an area for development, through which a greater understanding of the influence of oral QSMs on host cells could be achieved.

## 1. Introduction

The oral microbiome naturally serves to maintain an equilibrated ecosystem, facilitating the protection against disease development. This community of oral bacteria is enclosed in a self-produced extracellular polymeric matrix, forming oral biofilms known as plaque. The composition and delicate balance within this community influences the function of the oral microbiome. Oral diseases remain a major health problem worldwide, with the World Health Organisation (WHO) 2022 reporting that oral health neglect affects almost half of the world’s population [1]. Furthermore, oral diseases have been associated with other systemic diseases. In an effort to improve overall healthcare and well-being, a greater understanding of the environment within the oral cavity is essential in oral and dental research [2].

An important characteristic of biofilms is antimicrobial resistance (AMR), referring to the ability of microorganisms to withstand the effects of drugs that were designed to kill them or inhibit their growth. This constitutes a global health issue as infectious diseases, including oral diseases, caused by biofilms are difficult to treat and result in high morbidity. Examples are prosthetic joint infections, infective endocarditis and periodontitis [3]. Biofilms protect bacteria from the host immune system mainly through two mechanisms. First, the matrix acts as a physical barrier against antimicrobial reagents. Second, bacteria are able to exchange AMR genes, such as *mecA*, *vanA*, and *blaCTX-M* [4]. Biofilm communities reportedly are 10–1000 times more resistant to antimicrobial reagents than their planktonic counterparts [5]. Thus, it is crucial to explore innovative and sustainable alternatives for treatment through research endeavours.

Quorum sensing (QS) is a form of bacterial communication, regulating gene expression in response to cell density-dependent signalling molecules. For an oral biofilm to exist, QS is essential in coordinating bacterial behaviour in the establishment of complex functional communities [6]. Quorum sensing molecules (QSMs) have been found to influence oral biofilms in various ways, including growth, iron, and hemin uptake; stress-related gene expression; and virulence factor production [7]. Furthermore, QSMs have been discovered to indirectly influence host cells by shifting characteristics of an oral biofilm from a mutualistic to a pathology-associated state. This occurs through changes in species’ population ratios and/or virulence factor expressions, such as proteolytic activity, which are well-known indicators of the shift away from commensal biofilms [7,8].

The understanding of microbiology has vastly advanced over time. From the idea that bacteria exist as single cellular organisms to the understanding that biofilms hold actively coordinated multispecies communities, comes the current paradigm that cross-kingdom communication exists between bacteria and eukaryotic host cells [9]. From this, the suggestion that QSMs influence oral disease progression has emerged [10]. However, knowledge of direct oral QSM–host interaction is limited, and such studies could provide more insight into the cross-kingdom communication occurring during oral disease development. This review aims to explore the literature on oral QSM–host interaction and highlights areas of advancement. Developments in this field can open new avenues for targeting QS as a therapeutic method to treat oral diseases.

## 2. Common Oral Diseases and the Possible Role of Quorum Sensing

To understand the possible interactions between oral QSMs and the host response, it is first beneficial to recognise the cell configuration within the oral cavity and their role in oral disease development. The tissue and cell arrangement in the oral cavity is illustrated in Figure 1. According to the WHO, the following are classified as the most common oral diseases, estimated to affect 3.5 billion people [1].

Dental Caries (tooth decay)Periodontitis (gum diseases)Oral carcinoma (cancer)

Dental caries results from the metabolic processing of fermentable sugars into acids by oral bacteria, such as *Streptococcus mutans*, leading to the demineralisation of enamel and dentine. Three factors besides the presence of teeth determine the progression of dental caries: the oral biofilm dynamics, the oral environment, and time [11]. From this, it could be concluded that the synchronised increase in QSMs over time may facilitate the progression of dental caries.

Periodontitis is the dysregulated excessive inflammatory response of periodontal tissues towards a dysbiotic biofilm. Bacteria within plaque release virulence factors, leading to the recruitment of inflammatory cells [12]. This often leads to the loss of marginal gingival connective tissue and tight junctions between epithelial cells [13]. As QS and periodontitis are influenced by bacterial density, it could be hypothesised that there is a correlation between these two, which has become an area of recent interest.

Periodontitis is accompanied by an increase in epithelial migration, cell proliferation, and release of growth factors and inflammatory cytokines, all of which are closely associated with the development of oral cancers [11]. Currently, there is no direct correlation between QSMs and oral cancer development; however, QSMs present in the oral cavity have been discovered to upregulate factors associated with cancer development, as described in detail later on [14,15,16].

This review will explore if any of the discovered oral QSMs have an influence on the host response or the integrity of oral tissues, highlighting how QS may directly or indirectly contribute to the onset and progression of oral diseases.

## 3. Quorum Sensing in the Oral Cavity

Quorum sensing (QS) is a cellular signalling mechanism that allows bacteria to sense and respond to changes in cell population density through gene regulation. Bacteria in the oral cavity utilise quorum sensing to coordinate the expression of genes and adjust their collective behaviour based on the density of their immediate population [17]. QS follows the general mechanism of gene-activated signal production, signal release, signal accumulation, signal detection, and response, as illustrated in Figure 2. Initially, communication was thought to occur exclusively between Gram-positive and Gram-negative bacterial strains, respectively. However, autoinducer-2 (AI-2) has demonstrated the possibility to mediate cross-species communication between both domains [6].

## 4. Influence of Autoinducer Peptides on Host Cells

Autoinducer peptides (AIPs) are small oligopeptide signalling molecules secreted by Gram-positive bacteria. AIPs allow bacteria to communicate with each other and coordinate gene expression in response to changes in cell population density. By detecting the concentration of AIPs in their environment, bacteria can determine the local cell density and adjust their behaviours accordingly [18]. Receptors are usually transmembrane proteins that channel this signal intracellularly, and sensor histidine kinases transduce this signal to QS-regulated target genes [19]. In streptococci, 124 competence-stimulating peptide (CSP)-genes have been identified, affecting multiple pathways, including biofilm formation, acid tolerance, and virulence activity [20,21,22]. QS systems that have been described in Gram-positive oral bacteria include the CSP/ComABCDE and *com*X-inducing peptide (XIP)-ComRS signalling systems. A further PdrA/WGK system was recently identified in *S. mutans*, along with the fatty acid signalling molecule trans-2-decanoic acid [23,24].

Interestingly, host cell membrane proteins have been discovered to recognise bacterial CSP molecules. Specifically, these are bitter taste receptors (taste family 2 bitter receptor proteins; T2R), G protein-coupled receptors (GPCRs) that signal through Gα-mediated C-di-adenosine monophosphate (cAMP) decrease, and Gβγ activation of phospholipase C (Gβγ/PLC) leading to calcium release. The majority of T2Rs reside in the respiratory tract, with the oral cavity containing 25 variations of T2R. Several studies have detected genes for T2R in non-respiratory tissues, including the gastrointestinal tract, kidneys, and lymphocytes, suggesting that T2R may also have a chemosensory function in other organs. In oral tissues, T2Rs have been discovered to serve a diverse array of chemosensory functions, including innate immunity within the oral epithelium. Furthermore, T2Rs were shown to act as essential sensors towards bacterial QSMs [14,25,26].

Out of CSP-1, 2, and 3 produced by *S. mutans*, only CSP-1 induced a robust intracellular calcium mobilisation via the receptor complexes T2R14-Gβγ/PLC-β pathway within gingival epithelial cells (TIGK-hTERT). The immune response of this interaction was further examined with an enzyme-linked immunosorbent assay (ELISA), monitoring the expression of inflammatory cytokines. An increase in interleukin (IL) 6, IL-8, and tumour necrosis factor-α (TNF-α) was observed upon 18 h treatment of the gingival epithelial cells with 50 μM CSP-1 whilst IL-2, IL-4, and IL-10 were unaffected. CSP-1 was also observed to mediate NF-kB signalling. The regulation pathways leading to the observed characteristics remain an area for further research [26]. The activation of IL-8 and TNF-α by CSP-1 may be followed by an infiltration of immune cells, such as neutrophils and macrophages; however, only chemotactic immune cell testing can confirm this [27]. Investigating why the structure and chemical characteristics of CSP-2 and CSP-3 may not elicit a host response may hold the key towards a more comprehensive understanding of oral QSM-host interactions.

Medapati et al. further examined the interactions between T2R14 and *Staphylococcus aureus* and *S. mutans*. They found that oral epithelial T2R14 gene knockout significantly decreased the internalisation of *S. aureus* whilst *S. mutans* internalisation remained unaffected. T2R14 affected the cytoskeleton of oral epithelial cells, thereby inhibiting *S. aureus* internalisation. Interestingly, when oral epithelial cells were primed with *S. mutans* CSP-1, the growth of *S. aureus* was inhibited, but *S. mutans* growth was unaffected [26]. In support of this finding, recent work reports that bioactive molecules isolated from *S. mutans* biofilms demonstrate bactericidal effects on other commensal bacteria [28]. The results from the T2R14 study, in conjunction with the reported bactericidal effects of *S. mutans* biofilms indicate that the underlying mechanism may be attributable to QSMs [29].

Additionally, a study conducted by Kaur et al. explored the influence of dimethyl thiourea (DMTU), an identified potential inhibitor of ComA [30]. ComA is a protein involved in the CSP synthesis of some Gram-positive bacteria including *S. mutans* and is a transcription factor that plays a key role in regulating gene expression in response to cell density. In previous studies, Kaur et al. demonstrated that DMTU decreased biofilm formation of *S. mutans* [31]. Specifically in relation to caries development, DMTU down-regulated genes downstream of *comA* in *S. mutans* strains. These included the biofilm and virulence-related genes *comA*, *nlmC*, *immA*, *immB*, *bsmH*, *bsmI*, *comDE*, *comX*, and *comB*. The topical application of DMTU in Wistar rat models led to a significant decrease in biofilm formation and lowered the occurrence of dental caries compared to the control group. Moreover, the treatment with DMTU resulted in reduced levels of inflammatory markers in blood and liver samples of these animals. These findings suggest that DMTU may exhibit anti-biofilm and anti-inflammatory properties through ComA/CSP inhibition [30].

Further to the three studies described above, no other work has specifically evaluated the effects of CSPs present in the oral cavity on oral host cells. Nonetheless, other studies add value to this field. The influence of CSP from *Streptococcus mitis* on breast cancer cell invasion was investigated. Microscopic, transcriptomic, and chick chorioallantoic membrane (CAM) analysis demonstrated that CSP from *S. mitis* promoted tumour cell invasion and angiogenesis [15]. This new insight into the influence of oral QSMs on breast cancer progression may be translatable to oral epithelial carcinoma. However, to the best of our knowledge, no published research exists to investigate this hypothesis. If QSMs can be confirmed to have tumourigenic effects by multiple independent studies, antagonists to the identified QSMs may present opportunities in cancer prevention and therapy.

## 5. Influence of Oral *N*-Acyl Homoserine Lactones on Host Cells

### 5.1. Quorum Sensing Molecules Produced by Gram-Negative Oral Bacteria

*N*-Acyl homoserine lactones (AHLs), produced by AHL synthase enzymes, play a vital role as QSMs in numerous Gram-negative bacteria. AHLs are composed of a hydrophilic homoserine lactone ring and an amide group. They typically feature a saturated or unsaturated acyl chain attached at the C3 position, which can vary in length from 4 to 18 carbons (C4 to C18). AHL signalling function depends on the carbon chain configuration; for example, these chains may contain oxo- or hydroxyl groups, with the AHL structure reflecting the functional diversity. In the well-studied QS model organism *Pseudomonas aeruginosa*, *rhlI* is responsible for synthesising C4-HSL, increasing the production of virulence factors rhamnolipids. The N-(3-oxododecanoyl)-L-homoserine lactone (OdDHL), produced by *lasI*, is crucial in the bacterium’s cell growth cycle, with *P. aeruginosa* exhibiting higher levels in the early stationary phase and lower levels in the late stationary phase [32] AHL receptors are often cytoplasmic, and when activated, act as transcription factors for QS-regulated genes [6].

Due to the lack of oral AHLs discovered, it has been suggested that AHLs have a minor influence on oral biofilm development [33,34]. However, the current literature suggests this assumption may be due to underdeveloped detection technologies and methods rather than the absence of oral AHL molecules [33,34]. For example, a study conducted by Burgess et al. did not detect any AHLs in *Porphyromonas gingivalis* W50 using solvent-extracted supernatant and thin layer chromatography (TLC) with four different bacterial AHL biosensors. However, studies incorporating sophisticated chemical extraction and detection methods, such as high-performance liquid chromatography (HPLC), have now established the presence of AHLs in samples extracted from species present in oral biofilms, plaque, and saliva samples [35,36,37].

Additionally, Muras et al. found homologues of the AHL-synthase genes *hdtS* and *luxR*-type receptor in *P. gingivalis* W83 and ATCC 33277 strains [8,37]. These studies identified the presence of AHLs with a variety of carbon chain lengths through HPLC/mass spectrometry, including C8 AHL in saliva and teeth samples as well as C14, C18, OC8, and HC10 AHL in saliva samples.

### 5.2. Immune Responses to Oral AHLs

In the immunostimulatory analysis of AHLs, human Jurkat T lymphocytes were subjected to seven different AHLs at a concentration of 100 μM, which included the orally detected AHLs C8 and OC8. Among the tested AHLs, OdDHL, the well-defined QSM of *P. aeruginosa*, was the only AHL that led to apoptosis via the mitochondrial pathway [38]. Despite the fact that the orally present AHLs examined in this study did not elicit any stimulation response in immune cells, this work has discovered a possible mechanism whereby apoptosis-activating AHLs may interact with T lymphocytes.

Other studies have confirmed the possible inert characteristics of AHL molecules found in the oral cavity. C8 and C14 did not stimulate the production of inflammatory cytokines, IL-8, TNF-α, or IL-1β in THP-1 monocytes. Furthermore, activation of nuclear factor kappa B (NF-κB), a key signalling molecule involved in inflammatory immune responses, showed no changes when exposed to C8 or C14 AHLs [39]. Similarly, the inert nature of C8 and C14 was also reported in a range of non-oral cell lines, including EL4 T-lymphoblasts, DC2.4 dendritic cells, A20 mouse lymphoma cells, Raw264.7 mouse monocyte/macrophage-like cells, and THP-1 cells [40]. This study also investigated the mechanism of OdDHL–host interaction. OdDHL was discovered to incorporate into the mammalian plasma membrane, leading to lipid domain dissolution, resulting in apoptosis mediated by caspase 3 and caspase 8. This research holds interesting grounds for investigating the ability of other AHLs to interact with host cell membranes and to induce apoptosis in a similar fashion [40]. To consolidate the possible immunostimulatory effects of oral AHLs, different oral cell types, exposure methods, concentrations, and analysis techniques should be included in future studies.

### 5.3. Oral AHL Molecules Influence on Bitter Taste Receptors

The interaction between bitter taste receptors (T2Rs) and AHLs has also been an area of interest for analysing the influence of AHLs on host cells. The AHL–human interaction was studied through a unique sensory analysis method known as an electronic tongue, an analytical instrument containing chemical sensors capable of sensing and characterising chemical taste [41]. Following this, participants scored the bitterness of AHL compounds compared to known bitter standards, with identified T2R activation. Individual AHLs and mixtures scored higher bitterness than standards. These data indicate the presence of an interaction between T2R and AHLs. The study further analysed the functional characteristics of this observed interaction. Oral C8 AHL was found to activate T2R4, T2R14, and T2R20 on HEK293T human kidney epithelial cells with strong receptor potency, a measure of the magnitude of receptor activation. Characterisation of the amino acids in the interaction found that C8 AHL bound to similar sites on all three T2Rs, located on extracellular loop 2 [42]. The interaction between C8 AHL and T2Rs has provided a stepping stone towards further investigating the receptor activation by oral AHLs. Analysis methods such as a fluorescence resonance energy transfer (FRET) assay could be used to detect the oligomerisation state of membrane proteins in response to orally present AHLs.

After discovering an immune response activation of airway epithelial cells by AHL molecules through several T2Rs receptors, Freund et al. expanded their research to a broader range of specialised QSMs. Quinolones are naturally occurring plant and bacterial-derived molecules. Other than for their bactericidal properties, quinolones have been utilised by many species, including *P. aeruginosa*, as density-dependent QSMs. 2-heptyl-3-hydroxy-4-quinolone, known as *Pseudomonas* quinolone signal (PQS); 2,4-dihydroxyquinolone; and 4-hydroxy-2-heptylquinolone (HHQ) T2R activation within HEK293T cells, lung epithelial cell lines, and primary sinonasal cells was investigated. 

PQS at 10–100 μM activated T2R4, T2R16, and T2R38 in HEK293T cells; HHQ at 100 μM activated T2R14; and 2,4-dihydroxyquinolone had no effect. The activation of T2Rs was found to lead to an increased calcium induction, indicating activation of G-coupled T2Rs. This induction led to a decrease in stimulated cAMP levels in cell line cultures and primary airway cells; it was analysed with a combination of FRET-based protein biosensors and fluorescent indicator dyes. PQS and HHQ in primary sinonasal cells activated nitric oxide (NO) production to levels that were previously found to be bactericidal and to increase ciliary beating. Combined, the findings indicate that the presence of QSMs within the airways are utilised by host cells to activate an innate immune response via T2R, further leading to an increase in NO production, promoting clearance of pathogen-trapped mucus [25].

This study demonstrates the broadness of the interactions between QSMs and the host. It further indicates that bacterial quinolones, along with AHLs, trigger immune responses mediated by airway T2R. Furthermore, this study’s findings can potentially be translated to the oral cavity where the majority of T2R reside. Oral epithelial cell lines, such as H400 cells, as well as primary oral cells should be exposed to PQS and HHQ molecules in future research to understand the immune response induction they may have in the oral cavity.

### 5.4. Potential Carcinogenic Influence of AHLs

The theory that AHLs may have a role in carcinoma induction has been proposed by Sankar et al., hypothesising that AHLs might induce malignancy via the NF-κB signalling pathway through IκB phosphorylation and peroxisome proliferator-activated receptor (PPAR) inhibition. This theory is based on limited studies without any conducted tests [43]. Hence, in vitro analyses, such as oral epithelial cell proliferation investigations in the presence of oral AHLs and AHL inhibitors, are critical tests to evaluate this hypothesis. Furthermore, investigating changes in expression of crucial genes in the NF-κB signalling pathway would advance this area of research [44]. Additionally, carcinoma signalling pathways and the expression of cancer development genes, such as cyclin A, cyclin D1, cyclin-depended kinase 6, PRAD-1/CCND1, OSCCC should be analysed to aid the understanding of the possible influence of AHLs on carcinoma development [45].

## 6. Influence of the Universal QS Molecule Autoinducer-2 on Host Cells

### 6.1. AI-2 Communication in the Oral Cavity

The autoinducer-2 (AI-2)/LuxS system is considered the central QS system in oral bacteria, holding cross-species communication capability. AI-2 molecules are 4,5-dihydroxy-2,3-pentanedione (DPD) derivates formed from the enzymatic activity of the AI-2 synthases, encoded by the *luxS* gene. Farias et al. identified in vitro activity of the AI-2 system in 13 of 33 oral species. It has been shown that *S. mutans* species communicate with other streptococci utilising the LuxS system, affecting multiple virulence traits, including adhesion, cohesion, and acidic tolerance of *S. mutans* biofilms [46,47]. Adding AI-2 to *S. mutans* biofilms upregulates cariogenic, bacterial adhesion, cohesion, and biofilm formation genes. AI-2 was also found to regulate iron chelation and iron acquisition in *Aggregatibacter actinomycetemcomitans* and *P. gingivalis* [47].

### 6.2. Immune Activation by AI-2

*Fusobacterium nucleatum* is an oral commensal playing a crucial role in biofilm development and is associated with several human diseases, including periodontitis. *F. nucleatum* utilises AI-2 molecules in oral biofilm formation [17]. The influence of *F. nucleatum*’s AI-2 on macrophages has been investigated by Wu et al. A macrophage precursor cell line, U937, was exposed to *F. nucleatum’s* AI-2 molecules (50–400 μM), which were found to significantly increase macrophage migration. This phenomenon ceased in the presence of AI-2 inhibitor D-ribose. In addition, this study explored the impact of AI-2 molecules on macrophage immune responses by assessing markers for the pro-inflammatory M1-phenotype and the anti-inflammatory and tumour-promoting M2-phenotype. An ELISA analysis highlighted that M1 markers IL-8, TNF-α, and IL-1β were significantly increased at 400 μM AI-2 concentration. M2 marker IL-10 significantly decreased at AI-2 concentrations of 50 μM and 400 μM [16].

Despite its importance in biofilm formation, no clinical research into the concentrations of AI-2 found in oral disease has been undertaken to date. It could be assumed that disease-associated oral biofilms have a higher abundance of AI-2 molecules with increasing biomass; however, it is crucial to investigate and compare the levels of AI-2 in plaque associated oral diseases versus oral health [48,49,50]. Accordingly, macrophages may respond to increased AI-2 levels in these biofilms in an M1 macrophage pro-inflammatory manner. Hence, QSMs may indirectly contribute to increased tissue inflammation.

A recent study demonstrated that protein expression in macrophages was altered when co-cultured with purified *F. nucleatum* AI-2 molecules. Gene expression profiling highlighted 57 genes expressed at >1.3-fold change, of which 46 were upregulated and 11 downregulated. Further analysis revealed that the proteins overexpressed were associated with inflammatory factors and cytokine production, including TNFSF9, IL-1β, and C-C motif chemokines [51]. TNFSF9 is highly expressed in pancreatic cancer tissue and is associated with the M1 polarisation of macrophages [52]. This highlights the possible influence of *F. nucleatum*’s AI-2 on the progression of inflammatory diseases and cancer development.

### 6.3. Interactions between AI-2 QSMs and the Periodontium

*P. gingivalis* is a periodontal keystone pathogen due to its ability to trigger immune responses in the progression of periodontitis. To understand the relationship between AI-2 and periodontitis, *luxS* mutants of *P. gingivalis* have been developed [53]. These mutants showed decreased abilities to induce an immune response, with reduced expression of inflammatory cytokines IL-6, monocyte chemoattractant protein-1 and IL-1β produced by periodontal ligament (PDL) fibroblasts. This study suggests that AI-2 present in the oral cavity may induce an inflammatory response in PDL fibroblasts, aggravating periodontal disease development [54].

Additionally, it has been found that a range of mammalian cells can produce AI-2-mimicking molecules. *Vibrio harveyi* TL26, a reporter strain that is unable to synthesise AI-2 and does not respond to AI-1, was exposed to growth media from CaCo-2, Hela, A59, Jurkat E6-1, and U937 cells. Cells were identified to produce AI-2 homologues as luminescence was activated in the reporter strain, which only responds in this manner in the presence of AI-2 molecules. It was identified that AI-2 mimicking molecules are utilised by host cells to communicate with resident microorganisms. Although the investigated cell lines were not oral epithelial cells, these may present similar characteristics. Therefore, studies confirming the described cross-communication assay with oral epithelial cells are required [55].

## 7. Conclusions

The current literature presents QSM–host interactions as a possible novel pathway to treat oral diseases. However, this can only be achieved with a comprehensive understanding of QSMs within the oral cavity. QSMs from crucial Gram-positive bacteria have been well-defined. QSM CSP-1 produced by *S. mutans* was found to interact with T2R, activating NF-kB signalling and remodelling the cytoskeleton. Gram-positive QSMs influence inflammatory responses and may therefore facilitate the progression of periodontal diseases. Nonetheless, this interaction needs to be further evaluated using specific oral cell cultures and through clinical sampling. AI-2 in the oral cavity activates M1 macrophage characteristics known to have a pro-inflammatory role. AI-2 from *P. gingivalis* was found to elicit an inflammatory response in PDL fibroblasts. The QSM in this investigation was extracted from a key oral pathogen, and the cell line used was highly specific to the oral cavity; hence, this study paves the way for understanding oral QSM–host interactions.

AHL detection and influence within the oral cavity remains an area of limited findings. Further research on AHLs in the oral cavity is critical in understanding their influence on host cells. The current literature presents chemically inactive in vitro characteristics of two oral AHLs (C8 and C14); however, this has not been explored on oral cells. It is fundamental to first consolidate the findings of QSMs found in the oral cavity. This can be achieved through advancements in sensitive chemical analysis techniques and technology, which may include nuclear magnetic resonance spectroscopy, HPLC-MS and Fourier-transform ion cyclotron resonance (FT-ICR-MS) [56].

Studies incorporating oral epithelial cells, which are the primary cell type in contact with the oral environment, or other cells of the periodontium, would provide specific information on the influence of QSMs on oral infectious-inflammatory diseases. In addition, extracting QSMs from oral biofilms would lead to a better understanding of the specific QSMs produced and their association with symbiotic versus dysbiotic biofilms. Furthermore, researching the immune-activating effects of specific QSMs on oral cells including immune cells may reveal pathways of disease pathogenesis. The results from such investigations could subsequently lead to the identification of signalling cascades and genes influenced by QSMs, potentially serving as targets for QSM modulators. Examples include naturally occurring substances, such as cinnamaldehyde and curcumin as well as AHL-targeting enzymes (AHL lactonases and acylases) [57,58,59]. The combination of microbiology, human cell biology, and chemistry in the field of QS research can lead to clinical applications aimed at enhancing oral health.

## Figures and Tables

**Figure 1 microorganisms-11-01688-f001:**
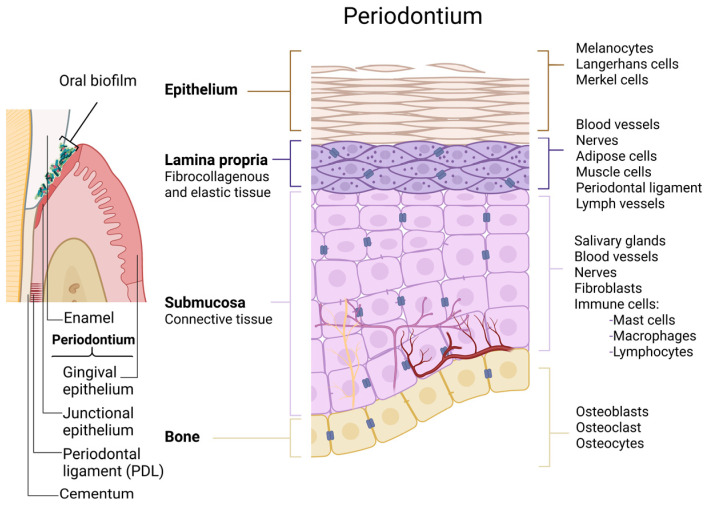
Schematic illustration of the organisation of oral tissues and residing host cells. The periodontium is composed of three functionally distinct tissue types, holding different compositions and arrangements of cells. Oral biofilms accumulate on the surface of both hard (teeth) and soft tissues (created in Biorender.Com).

**Figure 2 microorganisms-11-01688-f002:**
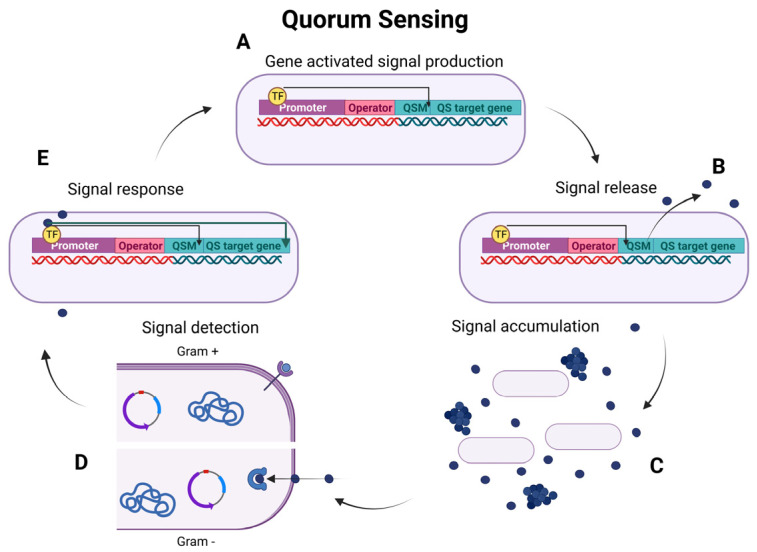
The general mechanism of QS, a cell density-dependent signalling mechanism in bacteria. TF: transcription factor. (**A**) Bacteria communicate via signalling molecules produced by QSM genes. QSM genes are expressed at low levels in low-density environments. (**B**) QSMs are released into the environment. (**C**) When a high cell density is reached, QSMs accumulate in the environment. (**D**) The accumulated QSMs are detected by neighbouring bacterial cells. In Gram–positive bacteria, QSMs are detected by membrane receptors transducing the signal intracellularly. In Gram–negative bacteria, QSMs are often membrane-permeable, and receptors are located intracellularly. (**E**) The detected QSMs activate the expression of various QS-controlled physiological genes in addition to the increased expression of QSM, leading to a positive feedback mechanism in the bacterial population (created in Biorender.com).

## Data Availability

All data presented within manuscript.

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
