# Peer review of "Quorum Sensing in Oral Biofilms: Influence on Host Cells"

_microorganisms, 2023, doi:10.3390/microorganisms11071688_

Round 1
Reviewer 1 Report
The paper presents a mature and solid review on the influence of quorum sensing molecules on human cells in the oral cavity. I have no major concerns, but recommend that the abstract should be modified to contain a synthesis of what is known from recent literature, rather than particular facts.
Other comments:
Line 16. AI-2 needs to be clarified.
Line 19. Fix double that.
Line 39 If I am not mistaken, "configure" refers to human activity. Please find another word.
Check your Ref. 2. The citation available at the linked page differs from what is on your reference list.
Section 2 is entitled "The role of host cells in the progression of oral diseases". However, it mainly discusses the implications of QS. Is it a good solution to rename the section as "Most common oral diseases"?
Line 85. has been found to induce --> is accompanied by
Figure 1. The line from the label "Cementum" to the image is lost. Label the enamel. Change the label "Peridontium" to "Periodontium" and move it down so that it does not overlap the image. Change macrophage to macrophages. Check the position of the label "Salivary glands". I am not a histologist, but found the following: "(...) small salivary glands (labial salivary glands) are present in the submucosa." [https://histology.medicine.umich.edu/resources/oral-cavity-salivary-glands].
Line 72. "interaction of fermentable sugars and microorganisms". Find a better phrase.
Line 74. Is the presence of the tooth surface a meaningful variable?
Lines 76-77. "no studies have explored this". Kaur and coauthors employed 1,3-di-m-tolylurea to inhibit an ABC transporter responsible for QSM export. It resulted in prevention of caries in rat model [PMID: 28748175]. Please check their study and decide if it is relevant to your statement and the provided evidence is satisfactory.
Lines 95-96. Check the description of the QS mechanism. If a "gene-activated signal production" takes place, cite an original publication.
Figure 2. In the caption, briefly describe each depicted step. For clarity, label parts of the figure with letters.
Lines 103 and 104; 159 and 161; 207 and 208. Remove abbreviation from section title and add explanation on the first line of a paragraph.
Minor English corrections are required
Author Response
Dear Reviewer 1,
Thank you for taking your time to look over our manuscript, we have attached our described changes in the attached word document.
Kind Regards,
Malee

Reviewer 2 Report
Dear Editors of Microorganisms Journal
I trust you are well. Herewith kindly receive my comments regarding the manuscript entitled “Quorum sensing in oral biofilms: influence on host cells”.
Kind regards
Musa Marimani

Please sort out the grammatical and spelling errors in the manuscript
Author Response
Dear Reviewer 2,
Thank you for taking the time to review our manuscript, we have attached a word document describing our amendments, from the help of your advice.
Kind Regards,
Malee

Round 2
Reviewer 2 Report
I am satisfied with the revised manuscript.
Thanks